# Adjusting the Structure of β-Cyclodextrin to Improve Complexation of Anthraquinone-Derived Drugs

**DOI:** 10.3390/molecules26237205

**Published:** 2021-11-27

**Authors:** Agata Krzak, Olga Swiech, Maciej Majdecki, Piotr Garbacz, Paulina Gwardys, Renata Bilewicz

**Affiliations:** 1Faculty of Chemistry, University of Warsaw, Pasteura 1, 02093 Warsaw, Poland; agata.krzak@chem.uw.edu.pl (A.K.); pgarbacz@chem.uw.edu.pl (P.G.); p.gwardys2@student.uw.edu.pl (P.G.); 2Faculty of Chemistry, Biological and Chemical Research Centre, University of Warsaw, Żwirki i Wigury 101, 02089 Warsaw, Poland; 3Institute of Organic Chemistry, Polish Academy of Sciences, Kasprzaka 44/52, 01224 Warsaw, Poland; mmajdecki@icho.edu.pl

**Keywords:** cyclodextrins, anthraquinone-2-sulfonic acid, anthraquinone-2-carboxylic acid, daunorubicin, association constant, solubility, inclusion complex

## Abstract

β-Cyclodextrin (CD) derivatives containing an aromatic triazole ring were studied as potential carriers of the following drugs containing an anthraquinone moiety: anthraquinone-2-sulfonic acid (AQ2S); anthraquinone-2-carboxylic acid (AQ2CA); and a common anthracycline, daunorubicin (DNR). UV-Vis and voltammetry measurements were carried out to determine the solubilities and association constants of the complexes formed, and the results revealed the unique properties of the chosen CDs as effective pH-dependent drug complexing agents. The association constants of the drug complexes with the CDs containing a triazole and lipoic acid (βCDLip) or galactosamine (βCDGAL), were significantly larger than that of the native βCD. The AQ2CA and AQ2S drugs were poorly soluble, and their solubilities increased as a result of complex formation with βCDLip and βCDGAL ligands. AQ2CA and AQ2S are negatively charged at pH 7.4. Therefore, they were less prone to form an inclusion complex with the hydrophobic CD cavity than at pH 3 (characteristic of gastric juices) when protonated. The βCDTriazole and βCDGAL ligands were found to form weaker inclusion complexes with the positively charged drug DNR at an acidic pH (pH 5.5) than in a neutral medium (pH 7.4) in which the drug dissociates to its neutral, uncharged form. This pH dependence is favorable for antitumor applications.

## 1. Introduction

Anthraquinone (AQ) derivatives constitute a large and diverse group, many of which have therapeutic properties and are used as chemotherapeutic [1,2], antiviral [3], immune boosting [4], or anti-inflammatory agents [5]. They are considered to be promising scaffolds for the development of antiviral drugs against SARS-CoV-2 [6,7]. They are also used as laxatives [8] for the treatment of malaria [9,10] and multiple sclerosis [11]. 6-Methyl-1,3,8-trihydroxyanthraquinone (emodine) has been investigated for the treatment of neurodegenerative diseases, and it has been reported to inhibit the pathological aggregation of tau protein. It was shown to protect against beta-amyloid-induced or H_2_O_2_-induced cortical neuronal deaths [12,13,14,15]. The synthetic quinone anthraquinone-2-sulfonic acid (AQ2S, Figure 1) studied in this work has an especially strong protective effect toward primary neurons, and the neuroprotective mechanisms of AQ2S are related to caspase inhibition [16]. Another drug, anthraquinone-2-carboxylic acid (9,10-dihydro-9,10-dioxo-2-anthracenecarboxylic acid (AQ2CA, Figure 1), has been shown to alleviate various inflammatory and pain symptoms, including EtOH/HCl and acetylsalicylic acid (ASA) gastritis, as well as inhibit the expression of inflammatory genes; it also acts as a potent anti-inflammatory ingredient in vivo, contributing to the regulation of the immune system [17].

Anthracyclines represent a distinct group of anthraquinone derivatives used to treat various types of cancer. The most popular representatives of anthracyclines, daunorubicin (DNR, Figure 1) and doxorubicin, are among the most effective anticancer drugs so far. They are used, among others, in the treatment of acute lymphocytic and myeloid leukemias, lymphomas, and bladder, breast, and brain cancer [18,19,20]. The cytostatic and cytotoxic effects of anthracyclines are explained by different mechanisms, including their interaction with topoisomerase II, which promotes growth arrest and apoptotic death of the tumor cells. It is also known that the anthraquinone moiety in anthracyclines intercalates between adjacent DNA base pairs, which leads to the inhibition of DNA and RNA synthesis, especially in highly replicating cells [21].

Despite the wide range of applications of anthraquinone-based pharmaceuticals, these drugs also have serious limitations as they have negative side effects during therapy [22]. They have the ability to generate reactive oxygen species (ROS) in the presence of cytochrome P450 reductase, NADH dehydrogenase, and xanthine oxidase (Appendix A). The excess ROS cannot be detoxified, which causes oxidative stress, DNA damage, and lipid peroxidation, thereby triggering cell apoptosis. Whereas this is a minor process for cancer cells, ROS also damages healthy tissues in the body. The most dangerous side effect of anthracyclines is cardiotoxicity [23,24].

Another disadvantage of anthraquinone-based drugs is their limited solubility in aqueous solutions [25]. The presence of hydrophilic substituents such as sugars may improve solubility; however, the therapeutic properties of these drugs are also changed. The solubility depends on the pH of the solution, leading to reduced absorption of drugs, and thus reduced bioavailability and therapeutic effect [26].

These negative properties of anthraquinone derivatives, i.e., poor solubility and the production of reactive oxygen species, can be improved by encapsulating the drug molecules in carriers [27,28], for example, binding them in the cavities of cyclic oligosaccharides, i.e., cyclodextrins (CD). The CD structure with its hydrophobic cavity allows the formation of an inclusion complex with hydrophobic drugs (Figure 2). Additionally, the presence of free hydroxyl groups outside the oligosaccharide ring allows CDs to be modified with substituents that influence the strength of drug binding [29,30]. In our previous work, we proved that properly modified cyclodextrins can increase the stability and solubility of poorly soluble drugs, based on the example of temozolomide [31].

In this work, we show that by using appropriate CD derivatives as carriers of anthraquinone-based drugs (Figure 2) we can both improve the solubility of the drug and increase the strength of the complex with the drug, paving the way to a reduction in unwanted side effects of these drugs connected with ROS production, as well as membrane permeability and solubility limitations. Moreover, NMR measurements for investigation of the βCD complexes binding site were performed and the pKa values for three cyclodextrin derivatives containing the triazole linker were determined. The presence of different kinds of substituents on the side chain of the triazole ring has been shown to have no effect on the pKa value of triazole-modified cyclodextrin and the pKa value is 4.5. We study three different guests (drugs) each of them containing an anthraquinone moiety. Depending on the pH, these drugs occur in different forms (cationic, anionic, or neutral); therefore, our aim was to establish how different pH values and the resulting charge of the drug affect its interaction with the cyclodextrin derivative. Our studies reveal that the form of the drug does matters, i.e., the charge of the drug and of the ligand are crucial, since the presence of electrostatic interactions between oppositely charged guest and host strengthens the complexes with cyclodextrin derivatives containing the triazole linker.

## 2. Results and Discussion

### 2.1. Determination of pKa Values of CD Derivatives

Due to the fact that the triazole linker in CD derivatives contains a basic nitrogen atom that can be protonated at an acidic pH, we determined the value of pKa by UV-Vis spectroscopy. The spectra of βCDGAL, βCDLip, and βCDTriazol, at the pH range between 2.7 and 9.5, are presented on Figure 3. The dependence of the absorbance of the peak maximum at wavelengths in the range of 210–220 on the pH of the buffer is presented in the figures. According to the equation pKa = [CDH^+^][H^+^]/[CD] for equal concentrations of the protonated and deprotonated forms, pKa = pH; the pKa value, for all three derivatives containing the triazole linker, was determined to be 4.5. Changing the CD substituent does not affect the acid dissociation constant of the triazole linker. A pKa value of 4.5 means that, at a pH below the pKa value, the CD derivative is usually in protonated form, and its additional positive charge in the triazole ring may affect the strength of the drug binding with the CD derivatives. For the amine derivative CD, the pKa value is available in the literature, and its value equal to 8.84 [32] does not affect the strength of the binding of this cyclodextrin with the drug at the range of physiological pH we tested.

### 2.2. Tuning β-Cyclodextrin Structure to Improve the Solubilities of AQ2CA and AQ2S Drugs

Anthraquinone-2-carboxylic acid (AQ2CA) is a poorly water-soluble drug (S_AQ2CA in water_ = 3.98 × 10^−5^ M). The solubility of AQ2CA in Britton–Robinson buffer, pH 7.4, is 1.74 × 10^−3^ M, while that in acidic solution (Britton–Robinson buffer, pH 3.0) is significantly lower (S_AQ2CA_ = 1.43 × 10^−6^ M). An improvement in the aqueous solubility of AQ2CA would improve the biological availability of the drug. Such a drug solubilization effect was achieved via complexation using appropriate CD derivatives. The phase solubility method developed by Higuchi and Connors was employed to quantify the solubilization ability of cyclodextrins [33]. The solubility values of AQ2CA in water, BR buffer at pH 7.4 (pH corresponding to the physiological conditions), and BR buffer at pH 3.0, with the addition of CDs was determined. The pH values characteristic for bodily fluids (pH 7.4) and for gastric juices (pH 3) were chosen for these measurements. The solubility diagrams (Figure 4) show that the solubility of anthraquinone-2-carboxylic acid increases linearly with an increase in cyclodextrin concentration (correlation coefficient > 0.99) up to 3 mM. According to the Higuchi and Connors classification [33], the phase solubility diagrams for anthraquinone-2-carboxylic acid with various cyclodextrin concentrations can be classified as A_L_ type. The diagrams, with linear correlation and a slope lower than 1, are characteristic for 1:1 complexation between the guest (AQ2CA) and host (cyclodextrin) molecules, suggesting that a good-solubility AQ2CA–CD complex was formed in the solution. The most spectacular increase in solubility was obtained in the presence of βCDLip and βCDGAL (almost tenfold increase in solubility at pH 3.0).

The association constant of the AQ2CA–cyclodextrin complex (1:1) was calculated from the linear plot of the phase solubility diagrams using Equation (1). The lines are started at the point of initial drug solubility, meaning, i.e., without the addition of cyclodextrin. The values of K_1:1_, as well as the solubility increase and correlation coefficients of the phase solubility diagrams, are presented in Table 1.

According to the phase solubility diagrams (Figure 4) and the association constant (K_1:1_) values (Table 1), βCDLip and βCDGAL featured an enhanced solubilizing effect and tlarger association constants of the complexes formed with these ligands. Complexes with βCD and βCDamine were weaker, especially in neutral solutions. The stronger βCDLip and βCDGAL effect is related to the presence of an aromatic triazole ring in their structures, which contributes to the strengthening of the complex through proton-acceptor π–π interactions with the aromatic ring of the drug [34]. These results confirm the beneficial effect achieved thanks to appropriate modification of the cyclodextrin. Larger association constants in the proton-rich environment (pH 3.0) are connected with the form of the drug. At pH 3.0, AQ2CA is in a neutral (protonated) form, whereas, at pH 7.4, it is anionic. The lower association constants at pH 7.4 reflect the lower affinity of the charged drug molecule to the hydrophobic cavity of the cyclodextrin.

The association constants of the AQ2CA– and AQ2S–cyclodextrin complexes were also determined by voltammetry using the Osa method (Equation (2)) in solutions of pH 3.0 and 7.4. The cyclic voltammograms for AQ2CA and AQ2S recorded in the absence and presence of βCDGAL are shown in Figure 5.

The addition of cyclodextrin to the AQ2CA or AQ2S solution led to a decrease in the voltammetric peak currents ascribed to the smaller diffusion coefficient of all drug–cyclodextrin complexes as compared with that of the free drugs. The dependencies of I^2^_obs_ vs. (I^2^_drug_ − I^2^_obs_)/[CD] for the AQ2CA–CD and AQ2S–CD complexes at pH 7.4 and pH 3.0 obtained using the CV method are shown in Figure 6. The values of the association constants for all complexes at pH 7.4 and 3.0 are exhibited in Table 2.

The values of the association constants of the AQ2CA–cyclodextrin complexes obtained using the cyclic voltammetry method were consistent with those obtained by UV-Vis spectroscopy. Moreover, both AQ2CA and AQ2S formed more stable complexes with the modified βCDs than with the native βCD, and the largest association constants were also obtained at pH 7.4 with βCDGAL. As mentioned above, strong proton-acceptor π–π interactions between the triazole ring of the cyclodextrin side group and the aromatic ring of the drug molecule was responsible for the increased association constants of the complexes. The association constants also depended on pH for AQ2S, and the weaker binding at pH 7.4 can be understood in terms of the lower affinity of the charged form of this drug for the hydrophobic CD cavity. The formation of strong inclusion complexes would increase solubility, thus, facilitating the delivery of the AQ2S and AQ2CA drugs and preventing the encapsulated drug from generating reactive oxygen species.

### 2.3. Determination of the Association Constants of Daunorubicin–Cyclodextrin Inclusion Complexes at pH 7.4. and 5.5

Due to the fact that a cancer cell environment is more acidic (pH 5.5) than that of healthy cells (pH 7.4), the carrier should be pH sensitive to enable the release of the DNR drug. The βCDs with triazole in the side chain were, therefore, chosen to compare the association constants of DNR–cyclodextrin complexes at these two values of pH. The same Osa (Equation (2)) dependencies were used to follow the changes in DNR reduction current in the absence and presence of the CDs. The Osa dependencies of I^2^_obs_ vs. (I^2^_DNR_ − I^2^_obs_)/[βCDGAL] for the two selected pH values are shown in Figure 7.

The values in Table 3 show that modification of cyclodextrin with a triazole linker increased the association constants of DNR–CD complexes, as seen in the case of AQ2CA and AQ2S. Exploiting the additional interaction between the drug aromatic ring and the triazole linker of cyclodextrin is, therefore, a general method of increasing the stability of the complexes. DNR–βCD derivatives were also sensitive to the pH change from 5.5 to 7.4.

Interestingly, the situation was different in the case of AQ2CA and AQ2S compared to DNR. At a pH of 5.5, which is characteristic of a cancer cell environment, these cyclodextrins formed weaker complexes with DNR than at a physiological pH (pH 7.4). Smaller values of stability at a lower pH (5.5) can be explained by the fact that DNR is in its cationic form at this pH, since the pKa value for DNR is 7.48 [35]. At a higher pH, the fraction of the deprotonated (i.e., neutral) form of the drug is larger; hence, its binding to the hydrophobic cavity increases. Thus, the difference in the charges of the anthraquinone drugs DNR and AQ2CA (at acidic pH, DNR is positively charged, while AQ2CA is negatively charged) is the reason for the different properties of their complexes at acidic and neutral pH. It may be noted that, at pH 5.5, the proton-acceptor π–π interactions should be additionally weakened due to the interaction of protons with lone electron pairs present on the nitrogen atoms of the triazole linker.

Significantly high values of the association constants of DNR inclusion complexes with βCDGAL at both pH values confirmed the affinity of drug molecules for the cyclodextrin cavity. Moreover, the higher stability of the complex at a physiological pH than at a pH of 5.5 is promising in view of cancer therapies. At a higher pH, the carrier strongly encapsulates DNR, protecting it from oxygen-radical formation reactions, while, at a lower pH, corresponding to the cancer environment, the release of the drug from the complex becomes favorable.

Lower values of the association constants of anthracycline complexes with βCDTriazole as compared with βCDGAL may indicate some contribution of the self-inclusion complex between the side substituents of the βCDTriazole derivative and the cavity of this cyclodextrin.

### 2.4. Determination of the βCD Complexes Binding Site by Nuclear Magnetic Spectroscopy

The structures of the complexes of βCD and anthraquinone-derived drugs were investigated by nuclear Overhauser effect spectroscopy (NOESY). The representative ^1^H NOESY spectrum for the βCDGAL/AQ2S system is shown in Figure 8. The ^1^H NMR signals of βCDGAL assigned to the protons inside the cyclodextrin cavity are located at 4.9–5.3 ppm. These signals have cross-peaks with the signals of aromatic protons of AQ2S (7.8–8.5 ppm), which shows that AQ2S is placed inside of βCDGAL. The reference ^1^H NOESY spectrum of a sample containing only AQ2S shows that the cross-peaks at 8.1 ppm placed vertically should be attributed to the internal interactions within the AQ2S ligand. Therefore, the lack of four vertical cross-peaks in the case of the external protons of βCDGAL, i.e., not resolved multiplets at ~3.6 ppm and ~5 ppm, excludes the binding site of AQ2S on the external surface of βCDGAL. Analogous results were obtained for the AQ2CA/βCDGAL system (see the supporting information for further details, Appendix A).

## 3. Material and Methods

### 3.1. Chemicals and Reagents

All reagents were of high purity (≥97%). Anthraquinone-2-carboxylic acid (AQ2CA) and anthraquinone-2-sulfonic sodium salt (AQ2S) were purchased from Sigma-Aldrich (Steinheim, Germany) and used without further purification. Daunorubicin (DNR) hydrochloride salt was purchased from AK Scientific (Union City, CA, USA). β-Cyclodextrin (βCD) and 6-monodeoxy-6-monoamino-β-cyclodextrin hydrochloride (βCDamine) were obtained from Sigma-Aldrich. The synthesis of the lipoic acid (βCDLip) [29], triazole (βCDTriazole), and galactosamine (βCDGAL) derivatives of β-cyclodextrin [30] was performed as previously described. Other compounds were purchased from Sigma-Aldrich. Buffers were prepared using water from a Milli-Q ultrapure water system. The Britton–Robinson buffers (pH 7.4, 5.5, and 3.0) were prepared in the usual way via the addition of appropriate amounts of 0.2 M sodium hydroxide to a 0.04 M solution of orthophosphoric acid, acetic acid, and boric acid. The pH was controlled using a pH Meter E2 (Mettler Toledo). The ionic strength of the buffer was adjusted to 0.5 M with potassium chloride. All experiments were carried out at room temperature (25 ± 1 °C).

### 3.2. UV-Vis Spectroscopy

The UV-Vis measurements were performed using an Agilent Technologies Cary 60 UV-Vis Spectrophotometer (Santa Clara, CA, USA). All UV-Vis spectra were measured using quartz cuvettes with a 1 cm optical path length. For pKa determination the 0.2 mg/mL concentration of βCDTriazole, βCDGAL or βCDLip in Britton–Robinson buffer were used.

### 3.3. Voltammetry

The cyclic (CV) and square-wave (SWV) voltammetry measurements were carried out using an EC Epsilon potentiostat (BASI). The electrochemical cell was kept in a Faraday cage. All electrochemical experiments were performed using a three-electrode arrangement with a glassy carbon electrode (BASi, 3 mm diameter) as the working electrode, a platinum foil as the counter electrode, and a silver/silver chloride (Ag/AgCl) electrode (BASi) in a saturated solution of KCl as the reference electrode. Before each electrochemical experiment, the surface of the glassy carbon electrode was polished using 0.05 μm alumina powder on a Buehler polishing cloth. After polishing, to remove traces of alumina from the electrode surface, the working electrode was rinsed with copious amounts of Milli-Q ultrapure water (resistivity 18.2 MΩ·cm).

### 3.4. Phase Solubility Diagrams

Solubility studies of anthraquinone-2-carboxylic acid with three different cyclodextrins were carried out according to the Higuchi and Connors procedure [33]. The solutions of cyclodextrin in water and Britton–Robinson buffers (pH 7.4 and 5.5) were prepared in a concentration range of 0–3 mM. A constant amount of AQ2CA (3 mM) that exceeded its solubility was added into the cyclodextrin solution. Molar ratios of AQ2CA/CD were 1:0, 1:0.2, 1:0.4, 1:0.6, 1:0.8, and 1:1. Three samples of each molar ratio were prepared. The suspensions were shaken for 24 h at 25 °C, after which equilibrium was reached. After 24 h, the concentration of the dissolved drug was monitored by UV-Vis spectroscopy and plateaued; thus, the maximum dissolved drug was achieved. Subsequently, the samples were filtered using a 0.45 µm filter and appropriately diluted. The aliquots of solutions were assayed for AQ2CA again, by UV-Vis spectroscopy, in the 200–600 nm range with a maximum absorbance value for AQ2CA at 335 nm. The apparent association constant (K_1:1_) for the complexes formed was calculated from the slope of the phase solubility diagram and the solubility of AQ2CA in water, Britton–Robinson buffer at pH 7.4, and Britton–Robinson buffer at pH 3.0, at 25 °C. The association constants of the inclusion complexes were determined using the following equation:(1)$$K_{1:1}=\frac{\mathrm{Slope}}{S_{0}\left(1-\mathrm{Slope}\right)}$$
where K_1:1_ is the association constant and S_0_ is the solubility of AQ2CA in the absence of cyclodextrin. The slope was measured by UV-Vis spectroscopy from the initial straight-line part of the AQ2CA concentration vs. CD concentration plot. The uncertainty was determined as the standard deviation of the three values of the association constants obtained from three different repetitions of the given system.

### 3.5. NMR Measurements

^1^H NOESY NMR spectra were acquired using an AVANCE III Bruker 500 MHz spectrometer equipped with a cryoprobe at the magnetic field strength of 11.75 T and room temperature (298 K). The recorded NOESY spectra have a high signal-to-noise ratio. We did not notice any significant loss of the signal-to-noise ratio due to the possible unfavorable tumbling rate estimated from the molecular weight of the studied cyclodextrin derivatives. See also [36] for a broaden discussion of NMR pulse sequences utilizing the Overhauser effect. The recorded NMR signals of cyclodextrin were assigned to ^1^H nuclei based on the data reported in [30,37]. The samples were prepared by dissolution of an equimolar mixture of βCDGAL and a derivative of anthraquinone (AQ2S, AQ2CA, and DNR) in deuterium-enriched water (99.5% D, Sigma-Aldrich). The obtained concentrations of saturated solutions of βCDGAL complexes varied in the range from 1 (DNR) to 10 mM (AQ2S and AQ2CA). The ^1^H NMR chemical shifts were referenced to the residual signal of HDO (4.79 ppm).

### 3.6. Evaluation of the Association Constants of Drug–Cyclodextrin Inclusion Complexes by Cyclic and Square-Wave Voltammetry

For the cyclic voltammetry experiments, the concentration of AQ2CA and AQ2S was 2.5 × 10^−5^ M, whereas the concentrations of βCD, βCDamine, βCDLip, and βCDGAL varied from 2.5 × 10^−4^ to 1.08 × 10^−3^ M. Cyclic voltammetry (CV) was carried out at a scan rate of 100 mV·s^−1^. For the SWV measurements, the concentration of DNR was 1.0 × 10^−6^ M, while the concentrations of βCD, βCDTriazol, and βCDGAL were increased in the range from 1.0 × 10^−5^ to 6.7 × 10^−5^ M. Before each SWV run, the electrode was electrochemically cleaned by applying a negative potential (−1.0 V) for 40 s, as described by Mora et al. [38]. The potential was varied between −0.2 V and −0.9 V at a frequency of 25 Hz, with an amplitude of 25 mV and a step size of 2 mV. Prior to all electrochemical measurements, the buffer solutions were purged with purified argon for 15 min.

AQ2CA, AQ2S, and DNR are electroactive compounds. As a result of the potential change, they undergo reduction and oxidation reactions involving two electrons and two protons (Figure 1).

Cyclic and square-wave voltammetry reduction peak currents were employed for the calculation of the drug–cyclodextrin complex formation constants based on the Osa equation [39]:(2)$$I_{\mathrm{obs}}^{2}=\frac{\left(I_{\mathrm{drug}}^{2}-I_{\mathrm{obs}}^{2}\right)}{K_{1:1}\cdot \left[\mathrm{CD}\right]}+I_{\mathrm{drug}:\mathrm{CD}}^{2},$$
where I_obs_ is the observed reduction peak current of the quinone group of the drug; I_drug_ and I_drug:CD_ are the reduction peak currents for the free drug and inclusion complex, respectively; K_1:1_ is the complex formation constant; and [CD] is the concentration of cyclodextrin. The value of K_1:1_ was calculated from the slope of the linear plot of I_obs_^2^ vs. (I_drug_^2^ − I_obs_^2^)/[CD].

## 4. Conclusions

β-Cyclodextrin derivatives containing an aromatic triazole ring as the side group were demonstrated to be suitable carriers of drugs which contain an anthraquinone moiety. The spectroscopic and voltammetry measurements showed that AQ2CA and AQ2S form stable water inclusion complexes with cyclodextrin with 1:1 stoichiometries. The phase solubility diagrams for the AQ2CA–cyclodextrin inclusion complexes were classified as A_L_ type. The advantage of using the designed cyclodextrins to complex the drugs lies in the increased solubility of these poorly soluble drugs, allowing the delivery of larger doses of the drugs when encapsulated in the CD carriers. βCDLip and βCDGAL derivatives formed stronger inclusion complexes with AQ2CA and AQ2S than with native βCD, due to the aromatic triazole linker, which was involved in direct π–π interactions with these drugs. The AQ2CA and AQ2S drugs are anionic in neutral medium, whereas they are protonated at a low pH (i.e., neutral). This led to the observed sensitivity of their CD complexes to changes in pH. At a pH of 3.0 (characteristic of gastric juices), the association constants of complexes with the neutral form of the drug were much larger than those at a physiological pH (7.4), when the drugs were in their anionic form and showed a lower affinity for the hydrophobic cyclodextrin cavity. In addition, cyclodextrin derivatives containing a triazole linker are protonated at a pH below 4.5, which may result in stronger binding of the anionic drugs. This is important in view of the potential medical applications of these drugs complexed with βCDLip and βCDGAL derivatives.

βCDTriazole and βCDGAL derivatives also formed stronger inclusion complexes with anthracyclines as compared with native βCD, as shown for the example of DNR. Interestingly, stronger inclusion complexes with DNR were, however, formed at a higher pH (7.4) than at a more acidic pH, in contrast to AQ2CA. This was due to the positive charge of DNR at a lower pH, which was less prone to form inclusion complexes with the modified CDs. Such behavior is therapeutically promising since, at a higher pH, which corresponds to the physiological pH of normal cells, some of the drug molecules would be in their neutral form strongly bound to cyclodextrin. Accordingly, they would not be released from the carrier, and therefore would not undergo negative side reactions. At pH 5.5, characteristic for the cancer cell environment, the positively charged drug would be released from the carrier, becoming free to intercalate into the DNA helices of the cancer cell.

It can be concluded that cyclodextrins modified with an aromatic triazole ring, especially βCDGAL, are promising carriers of drugs containing an anthraquinone moiety. The formation of CD complexes with the anthraquinone group improves the solubility of the drug and, more importantly, can protect the drug from unwanted side reactions, such as the generation of reactive oxygen species, which impede the application of these highly effective drugs.

## Data Availability

Not applicable.

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
