# Peer review of "Adjusting the Structure of β-Cyclodextrin to Improve Complexation of Anthraquinone-Derived Drugs"

_molecules, 2021, doi:10.3390/molecules26237205_

Round 1
Reviewer 1 Report
The manuscript by Krzak et al. describes the use of beta cyclodextrin and functionalized derivatives to complex with anthroquinone-derived drugs. The synthesis of the cyclodextrins was previously published, but this work investigates the solubility changes by complexation of three anthroquinones with cyclodextrins by UV-vis spectroscopy and voltammetry.
The authors show a significant improvement in solubility with the triazole and galactosamine functionalized cyclodextrins and their rationale seems reasonable. A quick search of the literature shows additional methods, such as NMR and DSC, being used to show that these inclusion complexes are formed or that that improve solubility or toxicity in vitro.
In the methods, TMZ is mentioned, which seems to be in reference to another drug that has been complexed by cyclodextrins by these authors and others. I hope this is a typo, but it makes me question the validity of the work. The wavelengths for detection of all the AQ2CA, AQ2S and DNR should be given, not TMZ.
Is there a reason the BR buffer was chosen for these studies?
Other suggestions:
Pg. 1, line 12, Pg. 10, line 283 and Pg. 8, line 213: Cyclodextrin at the beginning of the sentence needs to be capitalized.
Scheme 1 and 3: I think it should be labeled as figures.
Scheme 1: The structure of DNR is awkward with the hashed wedge on the ring and the other substituents (OH and NH3) up or down.
Paragraphs on pg. 2: It seems that there needs to be a few more references about serious limitations of the anthraquinone-based pharmaceuticals and the cytostatic and cytotoxic effects of anthracyclines.
Figure 1 legend: The diamond needs to be colored in for the data with betaCDLip.
Pg. 5, line 125: values is in there twice
Pg. 6, line 150: for all of all complexes?
Pg. 8, line 213: purchased instead of purches
Ref. 31: Authors are missing
Author Response
Reply to the comments of REVIEWER 1
The authors show a significant improvement in solubility with the triazole and galactosamine functionalized cyclodextrins and their rationale seems reasonable. A quick search of the literature shows additional methods, such as NMR and DSC, being used to show that these inclusion complexes are formed or that that improve solubility or toxicity in vitro.
Our reply
In the revised version we extended the manuscript including the NMR experiments. In particular, we investigated the structure of bCD complexes by adding the 1H NOESY spectra for the bCDGAL/ AQ2S and AQ2CA/bCDGAL systems. The obtained results confirm that AQ2S and AQ2CA bind in the cavity of bCDGAL. We added as point 2.4 in the manuscript two paragraphs and a figure presenting the 1H NOESY spectrum of the bCDGAL/AQ2S system.
In the methods, TMZ is mentioned, which seems to be in reference to another drug that has been complexed by cyclodextrins by these authors and others. I hope this is a typo, but it makes me question the validity of the work. The wavelengths for detection of all the AQ2CA, AQ2S and DNR should be given, not TMZ.
Our reply
The drug name should be indeed AQ2CA not TMZ - corrected, thank you. Only AQ2CA solubility was determined using the UV-Vis method.
Is there a reason the BR buffer was chosen for these studies?
Our reply
Britton Robinson buffer was used because of its wide pH range (2-12). Thanks to this, it was possible to study the systems in the pH range from 3 to 7.4 without changing the buffer composition. The ionic strength of the buffer solution was kept constant.
Other suggestions:
Pg. 1, line 12, Pg. 10, line 283 and Pg. 8, line 213: Cyclodextrin at the beginning of the sentence needs to be capitalized.
Our reply
Corrected, thank you
Scheme 1 and 3: I think it should be labeled as figures.
Our reply
Schemes 1 and 3 are now labeled as Figures 1 and 2.
Scheme 1: The structure of DNR is awkward with the hashed wedge on the ring and the other substituents (OH and NH3) up or down.
Our reply
The structure of the DNR have been changed, thank you for your attention.
Paragraphs on pg. 2: It seems that there needs to be a few more references about serious limitations of the anthraquinone-based pharmaceuticals and the cytostatic and cytotoxic effects of anthracyclines.
Our reply
References 22 and 24 were added.
Figure 1 legend: The diamond needs to be colored in for the data with betaCDLip.
Change were introduced, thank you.
Pg. 5, line 125: values is in there twice
Corrected, thank you.
Pg. 6, line 150: for all of all complexes?
Our reply
Was:
The addition of cyclodextrin to the AQ2CA or AQ2S solution led to a decrease in the voltammetric peak currents ascribed to the smaller diffusion coefficient of the drug–cyclodextrin complex compared with that of the free drug.
Now is:
The addition of cyclodextrin to the AQ2CA or AQ2S solution led to a decrease in the voltammetric peak currents ascribed to the smaller diffusion coefficient of the all drug–cyclodextrin complexes compared with that of the free drugs.
Pg. 8, line 213: purchased instead of purches
Our reply
Corrected, thank you.
Ref. 31: Authors are missing
Our reply
The names were added.
Reviewer 2 Report
The present manuscript focus on the inclusion complex formation of native and chemically modified b-cyclodextrins with anthraquinone-derived drugs. The improved solubilities and complex stabilities are investigated with the aim to contribute the potential widespread applications of these pharmaceuticals. Although the paper is well-designed, some questions should be analyzed prior the publication.
The authors used Britton-Robinson buffer to mimic the pH of the body fluids and gastric juices. It is known that the ionic strength affects the weak interactions such as cyclodextrin – drug complex formation. There is a large difference between the ionic strength of the acidic and basic components of the BR buffer, therefore, the solutions of different pH will have different ionic strengths. Have the authors checked whether this effect was negligible in addition to the charged form of the drug what they analyzed?
The authors write that „The values of the stability constants of the AQ2CA-cyclodextrin complexes obtained by the cyclic voltammetry method are consistent with those obtained by UV-Vis spectroscopy.” How they explain the difference of the data of AQ2CA-bCDamine at pH 7.4 determinate at the two methods?
Other minor remarks:
The use of abbreviations is not consequent. See e.g. page 4, lines 90-109. It is advisable to define abbreviation of the given word at the first appearance and then consistently use the abbreviation from there.
The introduction support understanding the importance of the present investigations, however, in some cases it contains general information that is too detailed and unnecessary for the work. E.g. Scheme 2.
Suggest using the same number format of Ks values in the different tables (Tables 1, and 2 vs Table 3 – decimal vs scientific notation).
Compared to the discussion, the conclusion seems too long and too detailed, although it summarizes the results an understandable way.
Author Response
Reply to the comments of REVIEWER 2
The authors used Britton-Robinson buffer to mimic the pH of the body fluids and gastric juices. It is known that the ionic strength affects the weak interactions such as cyclodextrin – drug complex formation. There is a large difference between the ionic strength of the acidic and basic components of the BR buffer, therefore, the solutions of different pH will have different ionic strengths. Have the authors checked whether this effect was negligible in addition to the charged form of the drug what they analyzed?
Our reply
The ionic strength of the buffer was adjusted to 0.5 M with potassium chloride. The text in Materials and methods was corrected. Thank you.
The authors write that „The values of the stability constants of the AQ2CA-cyclodextrin complexes obtained by the cyclic voltammetry method are consistent with those obtained by UV-Vis spectroscopy.” How they explain the difference of the data of AQ2CA-bCDamine at pH 7.4 determinate at the two methods?
Our reply
We repeated the electrochemical measurements and the discrepancy remains. It may be connected with lower diffusion coefficient of the complex formed due to larger structures formation when the CDamine ligand is employed.
Other minor remarks:
The use of abbreviations is not consequent. See e.g. page 4, lines 90-109. It is advisable to define abbreviation of the given word at the first appearance and then consistently use the abbreviation from there.
Corrected, thank you.
The introduction support understanding the importance of the present investigations, however, in some cases it contains general information that is too detailed and unnecessary for the work. E.g. Scheme 2.
Our reply
Scheme 2 was moved to Supplementary Materials.
Suggest using the same number format of Ks values in the different tables (Tables 1, and 2 vs Table 3 – decimal vs scientific notation).
Our reply
Corrected, thank you.
Compared to the discussion, the conclusion seems too long and too detailed, although it summarizes the results an understandable way.
Our reply
We tried to make the conclusions more concise as suggested.
Reviewer 3 Report
In the abstract, authors should explain βCDLip and βCDGAL.
Line 34 : used as laxatives, [8] for the treatment of malaria
Figure 1 : Number of repetitions ? Authors must do experiments 3 times in order to confirm the value of Ks. Moreover, some lines do not pass through the measured origin. So I recommend to do 3 times each point and force to pass the line through the measured origin.
Line 115: Ks is for solubility constant. Here, authors measured an association constant which should be noted Ka or K1:1.
In general avoid the use of stability constant, best is to use association constant.
Table 1: how was the uncertainty determined?
"The stronger βCDLip and βCDGAL, effect is related to the presence of aromatic triazole ring in their structures, which contributes to the strengthening of the complex through proton-acceptor π-π interactions with the aromatic ring of the drug." These affirmation must be proved by NMR and MD studies (see and cite 10.1016/j.ijpharm.2020.119391).
Regarding Osa method, could authors add some information about terms of use (the range of drug and CD concentration and of association constant, ....).
Table 3 : be consistent on the presentation of results (don't use × 104)
the publication of a paper on cyclodextrin inclusion complexes based only on constant determination is no longer acceptable. Authors must add ITC experiments (to obtain thermodynamic parameters) and NMR measurements (to determine 3D structure).
Authors must determine pKa of CD derivatives.
No information about the temperature was present in the manuscript.
Author Response
Reply to the comments of REVIEWER 3
In the abstract, authors should explain βCDLip and βCDGAL.
Our reply
Explanation added, thank you.
Line 34 : used as laxatives, [8] for the treatment of malaria
Our reply
The comma has been removed.
Figure 1 : Number of repetitions ? Authors must do experiments 3 times in order to confirm the value of Ks. Moreover, some lines do not pass through the measured origin. So I recommend to do 3 times each point and force to pass the line through the measured origin.
Our reply
The number of repetition is 3. Figure 1 (now figure 4) has been enriched with SD and the lines started are started at the point of initial drug solubility meaning - without the addition of cyclodextrin.
Line 115: Ks is for solubility constant. Here, authors measured an association constant which should be noted Ka or K1:1.
Our reply
As suggested, Ks has been changed to K1:1 to distinguish from the acid dissociation constant Ka, which was also determined in the manuscript.
In general avoid the use of stability constant, best is to use association constant.
Our reply
Changed as suggested by the Reviewer.
Table 1: how was the uncertainty determined?
Our reply
The uncertainty was determined as the standard deviation of the three values of the association constants obtained from three different repetitions for a given system. Added on page 11, line 314
"The stronger βCDLip and βCDGAL, effect is related to the presence of aromatic triazole ring in their structures, which contributes to the strengthening of the complex through proton-acceptor π-π interactions with the aromatic ring of the drug." These affirmation must be proved by NMR and MD studies (see and cite 10.1016/j.ijpharm.2020.119391).
Our reply
We followed the suggestion of this Reviewer and added NMR experiments described now in in points 2.4, 3.4. Reference 35 was added. The new figure presenting the 1H NOESY spectrum of the bCDGAL/AQ2S system is also added. The structure of bCD complexes was confirmed by the 1H NOESY spectra of the bCDGAL/ AQ2S and AQ2CA/bCDGAL systems. The results obtained show that both AQ2S and AQ2CA bind in the cavity of bCDGAL which supports the conclusions provided in the original version of this manuscript.
Regarding Osa method, could authors add some information about terms of use (the range of drug and CD concentration and of association constant, ....).
Our reply
There are no limitations in using the convenient Osa approach as long as larger aggregation of the complexes are not formed. This would affect the D values and the peak current would decrease more than expected for the simple complex formation. The other limitation is of course that the drug should be electroactive.
Table 3 : be consistent on the presentation of results (don't use × 104)
Our reply
Corrected , thank you
the publication of a paper on cyclodextrin inclusion complexes based only on constant determination is no longer acceptable. Authors must add ITC experiments (to obtain thermodynamic parameters) and NMR measurements (to determine 3D structure).
Our reply
Following this suggestion the NMR experiments were added to the manuscript (see point 2.4 and 3.4) Sorry we do not have access to the ITC method but instead we propose the very convenient electrochemical method of Osa.
Authors must determine pKa of CD derivatives.
Our reply
pKa of the CD derivatives were determined and added (point 2.1).
No information about the temperature was present in the manuscript.
Our reply
Information on the temperature was added in Materials and methods, thank you.
Round 2
Reviewer 2 Report
The authors made effert to improve the quality of the manuscriot and they ansvered several questions, therefore I suggest publishing the paper in Molecules.
Author Response
Our Reply: In the previous paper we were interested only in solubility improvement of temozolomide. In this paper, in contrast to the previous article, NMR measurements were performed and pKa for three cyclodextrin derivatives containing the triazole linker were determined. The presence of different kinds of substituents on the side chain of the triazole ring has been shown to have no effect on the pKa of triazole-modified cyclodextrin in the physiological pH range and the pKa value is close to 4.5. This value allows the use of this type of derivatives primarily in oral medications. Importantly, in this work we studied three different guests (drugs) which contained an anthraquinone moiety. Depending on the pH, this drugs occur in different forms, cationic, anionic or neutral, therefore our aim was to check how different pH and the resulting drug charge affects its interaction with cyclodextrin derivatives. Our studies show that the form of the drug does indeed matter: dependence on the charge of the drug and of the ligand are crucial since the presence of electrostatic interactions between oppositely charged host and guest strengthens the complexes with cyclodextrin derivatives containing the triazole linker.
This information is included in section 1 (page 3, lines 86 and next):
In our previous work we proved that properly modified cyclodextrins can increase the stability and solubility of the poorly soluble drug which is an effective alkylating compound [31].
In this work, we show that, by using appropriate CD derivatives as carriers of anthraquinone-based drugs (figure 2), we can both improve the solubility of the drug and increase the strength of the complex with the drug, paving the way to a reduction in unwanted side-effects of these drugs connected with ROS production, as well as membrane permeability and solubility limitations. Moreover, NMR measurements for investigation of the βCD complexes binding site were performed and pKa for three cyclodextrin derivatives containing the triazole linker were determined. The presence of different kinds of substituents on the side chain of the triazole ring has been shown to have no effect on the pKa of triazole-modified cyclodextrin and the pKa value is 4.5. We study three different guests (drugs) each of them containing an anthraquinone moiety. Depending on the pH, these drugs occur in different forms (cationic, anionic or neutral), therefore, our aim was to establish how different pH and the resulting charge of the drug affects its interaction with the cyclodextrin derivative. Our studies reveal that the form of the drug does matters: the charge of the drug and of the ligand are crucial since the presence of electrostatic interactions between oppositely charged guest and host strengthens the complexes with cyclodextrin derivatives containing the triazole linker.
- Krzak, A.; Bilewicz, R. Voltammetric/UV–Vis study of temozolomide inclusion complexes with cyclodextrin derivatives. Bioelectrochemistry 2020, 136, 107587-107593.
Reviewer 3 Report
the authors have taken into account the comments made by reviewers. However there are two problematic points :
1) "The number of repetition is 3. Figure 1 (now figure 4) has been enriched with SD and the lines started are started at the point of initial drug solubility meaning - without the addition of cyclodextrin."
this is good but the values of association constant have not changed in table 1 while the equations of the lines have necessarily changed with this modification. what do the authors say?
2) "We followed the suggestion of this Reviewer and added NMR experiments described now in in points 2.4, 3.4. Reference 35 was added. The new figure presenting the 1H NOESY spectrum of the bCDGAL/AQ2S system is also added. The structure of bCD complexes was confirmed by the 1H NOESY spectra of the bCDGAL/ AQ2S and AQ2CA/bCDGAL systems. The results obtained show that both AQ2S and AQ2CA bind in the cavity of bCDGAL which supports the conclusions provided in the original version of this manuscript."
The use of NOESY for the determination of 3D structure is not the best solution. As you can seen in https://doi.org/10.1051/jcp:1999224 or https://doi.org/10.1007/978-94-011-4681-4_156, the best solution is to do TROESY or ROESY off resonance. So the authors must do TROESY or ROESY off resonance instead of NOESY. Moreover, they could propose a 3D model from NMR measurements.
Author Response
Reviewer 3:
the authors have taken into account the comments made by reviewers. However there are two problematic points:
1) "The number of repetition is 3. Figure 1 (now figure 4) has been enriched with SD and the lines started are started at the point of initial drug solubility meaning - without the addition of cyclodextrin."
this is good but the values of association constant have not changed in table 1 while the equations of the lines have necessarily changed with this modification. what do the authors say?
Our Reply: Association constant values collected in the table 1 were separately determined when the lines are started at the point of initial drug solubility meaning - without the addition of cyclodextrin. Unfortunately, in the program in which the plots for publication were created, in the presence of several dependencies on one plot, there was an inaccuracy (the lines have not started at the point of initial drug solubility meaning), which was corrected after the reviewer's remark. Therefore, the values collected in the table 1 were not changed, because the correct equations of the lines were used in their determination. Moreover, the values of the association constants were obtained not from one equations of the line, but from three equations of the line received from three different repetitions of the given system.
This information is included in section 2.2 (page 5, lines 146-148).
The association constant of the AQ2CA–cyclodextrin complex (1:1) was calculated from the linear plot of the phase solubility diagrams using Equation (1), when the lines are started at the point of initial drug solubility meaning - without the addition of cyclodextrin. The values of K1:1, as well as the solubility increase and correlation coefficients of the phase solubility diagrams, are presented in Table 1.
2) "We followed the suggestion of this Reviewer and added NMR experiments described now in points 2.4, 3.4. Reference 35 was added. The new figure presenting the 1H NOESY spectrum of the bCDGAL/AQ2S system is also added. The structure of bCD complexes was confirmed by the 1H NOESY spectra of the bCDGAL/ AQ2S and AQ2CA/bCDGAL systems. The results obtained show that both AQ2S and AQ2CA bind in the cavity of bCDGAL which supports the conclusions provided in the original version of this manuscript."
The use of NOESY for the determination of 3D structure is not the best solution. As you can seen in https://doi.org/10.1051/jcp:1999224 or https://doi.org/10.1007/978-94-011-4681-4_156, the best solution is to do TROESY or ROESY off resonance. So the authors must do TROESY or ROESY off resonance instead of NOESY. Moreover, they could propose a 3D model from NMR measurements.
Our Reply: We thank referee #3 for the interesting reference. Although the different dependence of the NOE signal intensity on molecular tumbling rate for NOESY and ROESY experiments is well-established in NMR literature, the correlation time is usually not known precisely in a particular system under study. For instance, in our case, we used a sensitive NMR cryoprobe that provided a high signal-to-noise ratio of our 1H NOESY spectra. Furthermore, the measurements performed by Pean et al. were conducted 20 years ago, so we took advantage of the development of NMR equipment. We agree that by applying off-resonance ROESY, we could gain experimental sensitivity, but this would allow us only to record the spectra in a shorter time, and it would not provide us with any new information relevant to our study.
This information is included in section 3.4 (page 12, lines 333-338).
1H NOESY NMR spectra were acquired using AVANCE III Bruker spectrometer equipped with a cryoprobe at the magnetic field strength of 11.75 T and room temperature (298 K). The recorded NOESY spectra have a high signal-to-noise ratio. We did not notice any significant loss of the signal-to-noise ratio due to the possible unfavorable tumbling rate estimated from the molecular weight of the studied cyclodextrin derivatives. See also Ref. [36] for broaden discussion of NMR pulse sequences utilizing the Overhauser effect.
[36] Péan, C.; Djedaïni-Pilard, F.; Perly, B. Reliable NMR Experiments for the Determination of the Structure of Cyclodextrin Inclusion Complexes in Solution. In: Proceedings of the Ninth International Symposium on Cyclodextrins, Labandeira, J.J.T., Vila-Jato, J.L. , Eds.; Springer, Dordrecht, 1999, pp. 659-662.